# Preparation, Characterization and Molecular Dynamics Simulation of Rutin–Cyclodextrin Inclusion Complexes

**DOI:** 10.3390/molecules28030955

**Published:** 2023-01-18

**Authors:** Chaokang Chang, Meng Song, Mingxing Ma, Jihong Song, Fengyi Cao, Qi Qin

**Affiliations:** School of Materials and Chemical Engineering, Zhongyuan University of Technology, Zhengzhou 450007, China

**Keywords:** rutin, cyclodextrin, complex, molecular docking, molecular dynamics simulation

## Abstract

Rutin is a natural flavonoid that carries out a variety of biological activities, but its application in medicine and food is limited by its water solubility. One of the classical methods used to enhance drug solubility is encapsulation with cyclodextrins. In this paper, the encapsulation of different cyclodextrins with rutin was investigated using a combination of experimental and simulation methods. Three inclusions of rutin/beta-cyclodextrin (β-CD), rutin/2-hydroxypropyl beta-cyclodextrin (HP-β-CD) and rutin/2,6-dimethyl beta-cyclodextrin (DM-β-CD) were prepared by the freeze-drying method, and the inclusions were analyzed using Fourier infrared spectroscopy (FTIR), X-ray diffraction analysis (XRD), differential scanning calorimetry (DSC) and ultraviolet–visible spectroscopy (UV) to characterize and demonstrate the formation of the inclusion complexes. Phase solubility studies showed that rutin formed a 1:1 stoichiometric inclusion complex and significantly increased its solubility. β-CD, HP-β-CD, DM-β-CD, rutin and the three inclusion complexes were modeled by using MS2018 and AutoDock 4.0, and molecular dynamics simulations were performed to calculate the solubility parameters, binding energies, mean square displacement (MSD), hydrogen bonding and radial distribution functions (RDF) after the equilibration of the systems. The results of simulation and experiment showed that rutin/DM-β-CD had the best encapsulation effect among the three cyclodextrin inclusion complexes.

## 1. Introduction

Rutin, a natural flavonoid, is a pale-yellow or pale-green powdered crystal [1], which is widely found in the leaves of rutaceae, tobacco leaves, dates, apricots, orange peel, tomatoes and buckwheat flowers; its molecular structure formula is shown in Figure 1a. Numerous studies have shown that rutin has rich pharmacological activities [2], including antioxidant [3], anti-inflammatory [4], anticancer [5] and antibacterial [6] effects, and can be used in the treatment of cardiovascular diseases [7]. However, rutin is almost insoluble in water, which can lead to a relatively low bioavailability and limit its application in food and drug products [8]. Therefore, to expand the application of rutin, it is necessary to improve the solubility of rutin in water.

The most widely used method to improve the water solubility of insoluble compounds is the cyclodextrin inclusion method [9]. Cyclodextrins have a cone-shaped structure with a cavity in the middle [10], as shown in Figure 1b, and they possess the property of “hydrophobic inside and hydrophilic outside” [11]. Among the common cyclodextrins, β-CD is the most widely used. Some studies have shown that β-CD can increase the solubility of soybean sapogenins [12]. The solubility of gallic acid was improved after the formation of inclusion complexes with β-CD [13]. β-CD, though widely used, is relatively insoluble in water. To counter this drawback, cyclodextrin derivatives have been used for inclusion in many studies [14]. For example, DM-β-CD formed a stable inclusion complex with dihydromyricetin [15], and HP-β-CD could improve the solubility and bioavailability of β-Lapachone (βLAP) and its derivative nor-β-Lapachone (NβL) [16]. The structural formulae of β-CD, HP-β-CD and DM-β-CD are shown in Figure 1c–e.

Various methods can be used to prepare cyclodextrin inclusion complexes, such as the freeze-drying method, the saturated aqueous solution method, the co-precipitation method [17], the anti-solvent method [18] and the grinding method. For example, Liu et al. prepared a dihydromyricetin/β-cyclodextrin inclusion complex using the freeze-drying method [19]. Omrani and Tehrani prepared a gallic acid/β-cyclodextrin inclusion complex by the solvent evaporation method [20]. Additionally, different methods can be used for the characterization of the inclusion complexes, including differential scanning calorimetry (DSC), Fourier infrared spectroscopy (FTIR), X-ray diffraction analysis (XRD) and visible–ultraviolet spectroscopy (UV) [21].

Computer simulations play an essential role in the study of clathrates and can obtain information that cannot be acquired by other methods [22]. For example, Roy et al. performed molecular docking after preparing an AMB/CD inclusion complex and obtained the most-stable configuration [23]. Pahari et al. obtained the most-stable structures of flavonols and HP-γ-CD inclusion complexes by molecular docking and studied the positions of guest molecules in the inclusion complexes [24]. Zhang et al. used molecular dynamics simulations to study the relationship between puerarin and daidzin structures of inclusion complexes with β-CD. Their computational results revealed that both puerarin and anddaidzin could induce a conformational change in β-CD [25]. Yan et al. studied a naringin/cyclodextrin inclusion complex through molecular dynamics simulations. The results of the binding energy indicated that the naringin/HP-β-CD complex was more stable than the naringin/β-CD complex [26]. Zhao et al. predicted the inclusion ratio, stability of inclusion complexes and microstructure of ethyl red and cyclodextrin using molecular docking and molecular dynamics simulations [27].

Some progress has been made in the study of rutin/cyclodextrin inclusion complexes. Nguyen et al. studied the formation and stability of four rutin/cyclodextrin inclusion complexes. Rutin and cyclodextrin could form a 1:1 inclusion complex, and the stability constants of the inclusion complexes of rutin with HP-β-CD and HP-γ-CD were greater [28]. β-CD, HP-β-CD and rutin inclusion complexes were prepared to improve their aqueous solubility, and the results showed that the solubility of the inclusion complexes was significantly improved. Additionally, the solubility of an HP-β-CD inclusion system was higher than β-CD [29]. Rutin inclusion complexes with β-CD were prepared using the co-precipitation method. The formation of the inclusion complexes was confirmed by DSC and XRD [30]. The results of the phase solubility study showed that the rutin/HP-β-CD stability constant ratio of rutin/β-CD was 19.5 M^−1^ higher [31]. Liu et al. prepared rutin/β-CD and rutin/DM-β-CD though freeze-drying, and the effect of the DM-β-CD in improving the water solubility of rutin was 146% higher than that of β-CD [32]. However, these studies primarily focused on the preparation and characterization of the inclusion complexes, and there were some limitations to studying the mechanisms at the molecular level. Molecular simulation can predict the binding effect of different cyclodextrins with rutin. For example, the solubility parameter can be used to predict the compatibility between mixtures, the binding energy (*E_binding_*) can predict the stability of the formed inclusion complex and the hydrogen bonding between the host and guest of an inclusion complex can be analyzed using the radial distribution function (RDF) [33,34,35]. These molecular simulation methods can analyze the interaction mechanisms between the host and guest of an inclusion complex, providing a reference and basis for selecting a solubilized carrier.

In this study, three rutin/cyclodextrin inclusion complexes were studied at different scales by molecular docking, molecular dynamics simulations and experiments. β-CD, DM-β-CD and HP-β-CD were selected to be encapsulated with rutin. Three inclusion complexes were prepared and systematically characterized. The interaction of rutin with the three cyclodextrins in clathrate was studied via molecular docking and molecular dynamics simulations. The experimental results were combined with the simulation results to compare the encapsulation effects of the three cyclodextrins with rutin.

## 2. Results and Discussion

### 2.1. Fourier Infrared Spectroscopy Analysis

The changes in the FTIR spectra, such as a shift in the characteristic bands, disappearance, a reduction in intensity or the appearance of new bands, provided a considerable amount of information related to the guest–CD interaction [27]. The FTIR spectra of the three cyclodextrins, rutin and the three inclusion compounds are shown in Figure 2.

In the FTIR spectrum of rutin shown in Figure 2a, the peak at 2939 cm^−1^ corresponds to the methylene stretching vibration absorption of rutinose, the peak at 1655 cm^−1^ corresponds to the carbonyl stretching vibration and the peak at 1399 cm^−1^ corresponds to the bending vibration of the methyl group. In the physical mixture of rutin/β-CD, many peaks corresponded to characteristic absorption peaks in the FTIR patterns of rutin, such as those at 2939 cm^−1^, 1655 cm^−1^, 1399 cm^−1^, etc. However, the plot of the physical mixture was not identical to the plot of rutin, which showed an absorption peak of β-CD at 1024 cm^−1^. Additionally, some peaks showed slight variations in intensity and position, which were caused by the presence of cyclodextrins. The characteristic peaks of β-CD were not obvious in the physical mixture, possibly because the positions of the characteristic peaks overlapped with or were similar to those of rutin. In the pattern of the rutin/β-CD inclusion complex, the disappearance of the characteristic peaks produced by rutin was clearly observed, and the pattern was basically consistent with that of β-CD, which indicated that, after the inclusion, small molecules of rutin entered the cavities of the β-CD, and the characteristic peaks of rutin were masked by the β-CD.

As shown in Figure 2b, in the spectrum of the physical mixture of HP-β-CD and rutin, the characteristic peaks of both HP-β-CD and rutin appeared, and comparing the spectrum of the mixture with that of HP-β-CD and rutin alone, it was observed that the peaks at 1653 cm^−1^ and 1598 cm^−1^ were produced by rutin, and the characteristic absorption peak of the HP-β-CD was at 1402 cm^−1^. The spectrum of the mixture was a simple superposition of the spectra of the two monomers. The characteristic peak produced by rutin completely disappeared in the spectrum of the inclusion complex, and the characteristic absorption peak of the mixture corresponded to that of HP-β-CD, indicating that the rutin molecule entered the cavity of the cyclodextrin molecule after the reaction, and the characteristic absorption peak was masked by the cyclodextrin, which proved the success of the inclusion reaction.

Figure 2c is similar to Figure 2b. The characteristic absorption peaks of both rutin and DM-β-CD were present in the spectrum of the mixture, and the characteristic absorption peaks produced by rutin did not appear in the spectrum of the inclusion complex, which proved that the rutin was successfully encapsulated into the DM-β-CD. The rutin and the three kinds of cyclodextrins successfully underwent an inclusion reaction, but the specific inclusion mode was unclear. This needed additional verification from the XRD and the alternative characterization methods.

### 2.2. XRD Diffraction Analysis

The XRD patterns of the three cyclodextrins, rutin the three physical mixtures and the three inclusions are shown in Figure 3.

As shown in Figure 3a, the XRD patterns of the pure rutin and β-CD showed the crystalline state of both the materials, so dense crystal diffraction peaks appeared in both the plots. In the XRD diffraction pattern of the physical mixture, there were diffraction peaks indicative of both β-CD and rutin, and the peaks between 15° and 30° were noticeably denser than those in the plots of the β-CD and rutin alone, indicating that the patterns of the rutin were simply superimposed. The morphology of the inclusion complex was different from that of the rutin and β-CD alone. Although some characteristic peaks of β-CD appeared, the crystal diffraction peaks were significantly sparse, and new diffraction peaks appeared at 17° and 27°, indicating the formation of a new material, i.e., the inclusion complex.

In Figure 3b, since the HP-β-CD was amorphous, the XRD pattern of the rutin had dense and sharp crystal diffraction peaks, while the pattern of the HP-β-CD was relatively smooth. The pattern of the physical mixture clearly illustrated the diffraction peaks of rutin on top of the cyclodextrin pattern, which was as a result of the superposition of the rutin and HP-β-CD patterns. In the pattern of the inclusion compound, the crystal diffraction peak of rutin completely disappeared, and the pattern was similar to the HP-β-CD pattern; this change in the solid state of the materials was attributed to the formation of amorphous inclusions [17], which indicated the success of the inclusion reaction. After the inclusion, it was assumed that the rutin changed its state and was no longer in the crystal form.

As shown in Figure 3c, the DM-β-CD was also in a non-crystalline form, so it was more similar to the HP-β-CD. The pattern of the mixture was also a simple superposition of the patterns of cyclodextrin and rutin, and the diffraction peak of rutin appeared on the basis of the cyclodextrin. Meanwhile, the diffraction peak of rutin in the inclusion complex completely disappeared, and the pattern was similar to that of the DM-β-CD, which indicated the successful inclusion of the rutin in the DM-β-CD.

### 2.3. Differential Scanning Calorimetry (DSC) Analysis

The DSC plots for the three cyclodextrins, rutin, the three physical mixtures and the three physical mixtures of the inclusion compounds are shown in Figure 4.

As shown in Figure 4a, a heat absorption peak appeared in the spectrum of the β-CD at 106 °C, which was caused by the loss of water due to the drying and subsequent weight loss of the β-CD. A heat absorption peak appeared in the spectrum of rutin at 166 °C, which was caused by the transformation of the rutin from a solid to a liquid state after heating and gradually reaching its melting point. The peak intensity of the melting point of rutin in the physical mixture was weakened and moved to 153 °C, which was possibly due to the inclusion reaction between some cyclodextrins and rutin during the heating process. In the plot of the inclusion complex, the characteristic peak of rutin completely disappeared, indicating that the inclusion of the rutin in the cyclodextrin was successful.

As shown in Figure 4b, the variation in the HP-β-CD at 239 °C corresponded to its melting point. In the physical mixture, a peak indicative of rutin appeared, but the intensity and position of the peak changed from 166 °C to 142 °C, which was caused by the inclusion reaction between part of the rutin and the cyclodextrin in a molten state during the heating process. The free rutin and cyclodextrin were no longer present in the inclusion complex, so the characteristic peaks of the substances did not appear.

In the DSC spectrum of DM-β-CD shown in Figure 4c, the phase transition at 242 °C was produced by gradually reaching the melting point. The reason for the change in the spectrum of the physical mixture was the same as that of the physical mixture shown in Figure 4a,b, which was also due to the inclusion of part of the rutin and the cyclodextrin during the heating process; as a result, the characteristic peak of rutin shifted from 166 °C to 185 °C. The characteristic peaks of the two substances no longer appeared in the inclusion pattern, indicating that the inclusion complex was a new substance. Thus, the inclusion was successful.

The thermal analysis proved that the thermal properties of the inclusion compounds changed. The melting point of the three inclusion complexes was higher than that of the free rutin, which indicated that the inclusion complex was more stable than the free rutin, but it could not be determined which of the three inclusion complexes was the most stable.

### 2.4. UV–Visible Spectral Analysis

#### 2.4.1. Detection Wavelength of Rutin

The UV scanning spectra of the β-CD, HP-β-CD, DM-β-CD and rutin are shown in Figure 5. Based on the UV spectra, we know that the maximum absorptions of rutin occurred at 265 nm and 355 nm, while the β-CD, HP-β-CD and DM-β-CD had no obvious UV absorption in the range of 200~600 nm. Thus, the β-CD, HP-β-CD and DM- β-CD had no interference with the maximum absorption of the rutin, and 265 nm was selected as the detection wavelength for the determination of the rutin content.

#### 2.4.2. Standard Curve of Rutin

We weighed an appropriate amount of rutin dissolved in 95% ethanol to prepare a series of solutions with concentrations of 0.02 mmol/L~0.09 mmol/L. The absorbance was measured at 256 nm, respectively. The measured data are shown in Table 1.

The standard curve of rutin drawn according to Table 1 is shown in Figure 6.

As shown in Figure 6, the linear relationship between the different concentrations of rutin and their absorbance was good, and the standard curve of rutin was calculated as y = 20.83333x + 0.06792 with a correlation coefficient R^2^ = 0.9833.

#### 2.4.3. Solubility Diagram of Rutin Phase

A series of aqueous cyclodextrin solutions were prepared in the concentration range of 2–10 mmol/L. A quantitative excess of rutin was added to pure water and the aqueous cyclodextrin solutions of different concentrations, respectively, and then sealed and stirred at room temperature for 48 h to bring the dissolution to equilibrium. The samples were then removed and filtered. Next, the filtrate was appropriately diluted, and the UV absorbance at 265 nm was measured (Table 2).

The data in Table 2 were substituted into the standard equation for rutin, and the phase solubility diagrams of the three cyclodextrins with rutin were plotted based on the calculated data, as shown in Figure 7.

The Benesi–Hildebrand method is a common strategy for estimating the stoichiometry and binding constants of inclusion complexes [21]. As shown in Figure 7, the phase solubility plots of all of the three cyclodextrins and rutin were linear and all belonged to the AL type, which proved that all of the three cyclodextrins were 1:1 encapsulated with rutin, which was consistent with the model of simulated partial molecules forming encapsulations according to a 1:1 docking. In the phase solubility diagram, the solubility concentration of rutin was ranked from largest to smallest in the following order: rutin/DM-β-CD system > rutin/HP-β-CD system > rutin/β-CD system. This was consistent with the solubility of the three cyclodextrins, which indicated that the better the solubility of cyclodextrins, the better the compatibility with rutin and the more favorable the inclusion reaction.

#### 2.4.4. Inclusion Stability Constants

The inclusion stability constant (*K_s_*) is an important parameter for measuring the stability of inclusion complexes. The larger the *K_s_* value is, the stronger the stability effect of the CD on the small guest molecules [31]. The calculation Equation (1) is as follows:(1)$$K_{s}=\frac{K}{S_{0}\left(1-K\right)}$$
where *K* is the slope of the standard equation in the phase solubility diagram, and *S*_0_ is the intercept of the standard equation for the inherent solubility of small guest molecules.

The stability constants of the three cyclodextrin–rutin inclusion complexes were calculated separately according to the phase solubility diagram and the equation of the inclusion stability constants: *K_S_* (rutin/β-CD) was 275.5 M^−1^, *K_S_* (rutin/HP-β-CD) was 442.5 M^−1^ and *K_S_* (rutin/DM-β-CD) was 1012.4 M^−1^. The larger the value of *K_S_*, the more stable the inclusion complex. Among the three inclusion complexes, rutin/DM-β-CD was the most stable followed by rutin/HP-β-CD.

### 2.5. Molecular Docking

Molecular docking was performed by using AutoDock 4.0. Cyclodextrin is a rigid receptor, and rutin is a semi-flexible ligand [32]. For the β-CD receptor, we created a docking box with 34 × 38 × 34 grid points, and the grid spacing was set at 0.375 Å. Based on the molecular size, the grid box size was set to 42 × 40 × 30 for the DM-β-CD receptor and 42 × 40 × 34 for the HP-β-CD receptor, and the remaining parameters were consistent with those selected for β-CD. The guest molecules were docked into the corresponding cyclodextrin cavities based on the Lamarckian genetic algorithm (LGA), and then 1000 conformational searches were performed. A cluster analysis was performed based on the energy of the docked conformations, and the optimal conformation was selected for further study [23]. The result of clustering analysis diagram is shown in Figure 8.

The corresponding most-stable conformations of the obtained inclusion complexes are shown in Figure 9.

As shown in Figure 9, the cavity of the β-CD was not large enough to fully encapsulate the whole rutin molecule, but the benzene ring end of the rutin appeared at the huge mouth end of the cyclodextrin. Two kinds of cyclodextrin derivatives could entirely encapsulate the rutin, indicating that the inclusion effect of the cyclodextrin derivatives and rutin was better than that of the β-CD.

### 2.6. Judgment of System Equilibrium

In order for the system to be in equilibrium, a series of simulations were required. There were two main bases for determining whether the system was in equilibrium: one was a temperature equilibrium and the other was an energy equilibrium, which either required a constant value or fluctuated up and down along a constant value. Figure 10a shows the temperature equilibrium diagrams of the three complexes at 298 K and 1000 ps of the NPT system synthesis, and Figure 10b–d show the energy equilibrium diagrams of the rutin/β-CD IC, rutin/HP-β-CD and rutin/DM-β-CD IC. There were up and down fluctuations in the temperature and energy of the three inclusions, but overall they were in dynamic equilibrium, indicating that the simulated system energy reached the equilibrium state.

### 2.7. Solubility Parameters Analysis

The solubility parameter δ is defined as the square root of the molecular cohesive energy density (CED). The calculation of δ is shown in Formula (2):(2)$$\delta ={\left(CED\right)}^{\frac{1}{2}}={\left(\frac{\Delta E}{V}\right)}^{\frac{1}{2}}={\left(\frac{\Delta H_{V}-RT}{V}\right)}^{\frac{1}{2}}\;$$
where Δ*E* is the cohesive energy of the system, Δ*H* is the heat of evaporation possessed by a mole, *V* is the corresponding molar volume and *RT* is the expansion work conducted by a mole of the system when vaporized. Solubility parameters can be used to determine whether different substances are compatible. If δ_A_ − δ_B_ < 1.0 (cal/cm^3^)^1/2^, A and B are compatible; if 1.0 (cal/cm^3^)^1/2^ < δ_A_ − δ_B_ < 3.4 (cal/cm^3^)^1/2^, A and B may be compatible; however, if δ_A_ − δ_B_ > 4.9 (cal/cm^3^)^1/2^, A and B are incompatible [36].

Table 3 presents the data obtained from the calculation.

According to the data in Table 3, the difference in solubility parameters between the three cyclodextrins and rutin was less than 3.4 (cal/cm^3^)^1/2^, and the compatibility order of DM-β-CD > HP-β-CD > β-CD was obtained according to the solubility parameters.

### 2.8. Binding Energy Analysis

The binding energy (*E_binding_*) is defined as the negative value of the interaction energy (*E_inter_*), which can reflect the interaction between two components [26]. The larger the *E_binding_* value, the stronger the interface interaction. If *E_binding_* is positive, the compatibility between two components is good. *E_binding_* can be calculated using Equation (3):(3)$$E_{binding}=-E_{inter}=-\left(E_{total}-E_{f}-E_{CD}\right)$$

*E_total_*, *E_f_* and *E_CD_* represent the total energy of the inclusion system, the energy of the small guest molecules of rutin and the energy of the cyclodextrin, respectively.

Based on the binding energies of the inclusion complex of β-CD with rutin, the inclusion complex of HP-β-CD with rutin and the inclusion complex of DM-β-CD with rutin averaged for frames 197~201, respectively, the obtained values were substituted into the formula, and the binding energies were calculated as shown in Table 4. According to the magnitude of the binding energy, the order of the intermolecular interaction forces was as follows: rutin/DM-β-CD IC > rutin/HP-β-CD IC > rutin/β-CD IC. This order corresponded to the results of the solubility parameters. The stability of the inclusion complexes proved by the binding energy results supplemented the deficiencies in the DSC analysis.

### 2.9. Mean Square Displacement Analysis

The MSD is calculated by the following Equation (4):(4)$$MSD=\langle {\left|r_{i}\left(t\right)-r_{i}\left(0\right)\right|}^{2}\rangle$$
where *r_i_*(0) and *r_i_*(*t*) are the positions of particle i at the initial time and time t, respectively, and the <> symbol indicates the average value of all particles. The mean square displacement simulations were performed for the free rutin molecules and the three cyclodextrin–rutin inclusions, respectively. The MSD diagram is shown in Figure 11.

As shown in Figure 11, the mean square displacement of the free rutin molecules increased continuously with the increase in the simulation steps, reaching its largest size at 700 ps, while the displacements of the three inclusions were significantly smaller after 700 ps, indicating the restriction of the rutin molecule after entering the cyclodextrin cavity. The MSD values of the three inclusions were different, and these were the smallest for rutin/β-CD, which might be related to the size of the cyclodextrin cavity. The rutin/DM-β-CD sample was smaller than rutin/HP-β-CD, but the two cyclodextrin cavities were similar in size, which might be because the rutin/DM-β-CD was more stable compared with the rutin/HP-β-CD. The results of the MSD analysis were consistent with the results of the binding energy analysis, which further supplemented the stability of the inclusion complex.

### 2.10. Hydrogen Bond Analysis of Cyclodextrin System

Among the three cyclodextrins, the water solubility of β-CD is the worst, and it is generally believed that the reason for its poor water solubility is related to its greater number of intramolecular hydrogen bonds [32]. The cubic periodic unit-cell model of the β-CD system containing only cyclodextrin molecules with a side length of 19.2 Å, the cubic periodic unit-cell model of the HP-β-CD system with a side length of 22.2 Å and the cubic periodic unit-cell model of the DM-β-CD system with a side length of 21.2 Å were constructed, and the hydrogen bonding in the β-CD, DM-β-CD and HP-β-CD systems were analyzed via kinetic simulations. The number and concentration of hydrogen bonds in the β-CD system were also analyzed via kinetic simulations, as shown in Figure 12.

A cell system of the three cyclodextrin monomers was established, and molecular dynamics simulations were performed. The number of hydrogen bonds in the system was counted, and the molar concentrations of hydrogen bonds (*C_HBs_*) were calculated based on Equation (5):(5)$$C_{HBs}=\frac{N_{HBs}/NA}{V}$$

In the above Equation, *N_HBS_* is the number of hydrogen bonds in the periodic cell, *NA* is Avogadro’s constant and *V* is the volume of the periodic cell.

The number of hydrogen bonds in the equilibrium system obtained after the molecular dynamics simulations was counted, and the molar concentrations of hydrogen bonds (*C_HBs_*) were calculated based on Equation (5); the results are listed in Table 5.

The blue dashed lines in Figure 12 represent hydrogen bonds, and it can be seen that the β-CD system and the HP-β-CD system have a large number of intra- and intermolecular hydrogen bonds formed, while the number of hydrogen bonds in the DM-β-CD system is relatively sparse. Based on the specific values in Table 5, the number of unit cell hydrogen bonds in the β-CD system is not significantly different from that in the HP-β-CD system and both are much higher than that in the DM-β-CD system. However, because of the effect of the unit cell volume, the β-CD system has the highest hydrogen bond concentration, which is one order of magnitude higher than the other two, followed by the HP-β-CD system, and the lowest is that of the DM-β-CD system. Stronger intramolecular hydrogen bonding may be a factor affecting its water solubility and its interaction with rutin. The hydrogen bonding concentrations of the three cyclodextrin systems were in the opposite order of solubility parameters and binding energy, proving that the hydrogen bonding in the cyclodextrin system affects the solubility of cyclodextrin and thus the interaction between the host and the guest and the inclusion effect.

### 2.11. Hydrogen Bond Predictive Analysis and Radial Distribution Function (RDF) of Inclusion Complexes

Intermolecular hydrogen bonds provide an essential contribution to the stability of inclusion complexes, second only to that of covalent bonds [37]. Based on the structural formulae of rutin and its three cyclodextrins, to predict the type of hydrogen bonds that may be formed between molecules, the atoms involved in the formation of intermolecular hydrogen bonds included the oxygen atoms of the rutin and cyclodextrins and the hydrogen atoms on the respective hydroxyl groups. The atoms with the same electronegativity on the same molecule were divided into one category, and the distribution of the atomic electronegativity of rutin and cyclodextrin is shown in Table 6.

According to Table 6, the possible types of intermolecular hydrogen bonds could be predicted, and a total of eight types of hydrogen bonds could be formed, which were type A (O1-H2), type B (O2-H2), type C (O3-H2), type D (O4-H2), type E (O5-H2), type F (O6-H2), type G (O7-H1) and type H (O8-H1).

The types of possible intermolecular hydrogen bond formations were predicted according to the structure of the host and the guest. Among the hydrogen atoms contained in the cyclodextrins and rutin, only the hydrogen atoms in the hydroxyl group were involved in the formation of intermolecular hydrogen bonds, and the type of possible intermolecular hydrogen bond formations could be predicted. However, the specific formation (or lack of) needed to be judged by the waterfall diagram obtained using the radial distribution function (RDF). The radial distribution function is defined as the relative probability of finding another atom at a distance r from one atom. It is used to analyze the distribution of atoms in a system and to predict the type of hydrogen bonds to be analyzed, as shown in Equation (6):(6)$$g\left(r\right)=\frac{dN}{4\rho \pi r^{2}dr}$$
where *dN* is the number of atoms occurring in the range from *r* to *r +* ∆*r*, and *ρ* denotes the average density of the system. When the value of *g*(*r*) is in the range of 2.5–3.0 Å, 3.0–5.0 Å and 5.0 Å, the intermolecular interactions are attributed to hydrogen bonds, strong van der Waals forces and weak van der Waals forces, respectively.

The radial distribution diagrams of the three inclusion systems are shown in Figure 13.

From Figure 13a, it can be seen that five types of hydrogen bonds (A, B, C, E and G) existed stably; three types of hydrogen bonds (A, B and G) existed stably in Figure 13b and four types of hydrogen bonds (A, B, C and G) existed stably in Figure 13c. According to the order of the peaks forming hydrogen bonds in the figure, from strong to weak: rutin/DM-β-CD > rutin/HP-β-CD > rutin/β-CD, and this order was consistent with the order of the binding energy.

## 3. Experimental Section

### 3.1. Materials and Instruments

β-CD (purity ≥ 98%) was obtained from Shanghai Aladdin Biochemical Technology Co., Ltd. (Shanghai, China). Rutin (purity ≥ 98%) was purchased from Shanghai Dibai Biotechnology Co., Ltd. (Shanghai, China). HP-β-CD (purity ≥ 97%) was obtained from Shanghai McLean Biotechnology Co., Ltd. (Shanghai, China). DM-β-CD (purity ≥ 98%) was purchased from Shanghai Dibai Biotechnology Co., Ltd. (Shanghai, China). Anhydrous ethanol (analytically pure) was obtained from Tianjin Fuyu Fine Chemical Co., Ltd. (Tianjin, China). The distilled water used in the study was produced by Zhongyuan University of Technology.

Other equipment used included a Nicolet IS50 Intelligent Fourier infrared spectrometer (Thermo Fisher Technologies, USA), an Ultima IV Powder Polycrystalline X-ray Diffractometer (Neki Corporation, Japan), a DSC822E thermal analyzer (Mettler Toledo, Switzerland), a UV1800PC UV–visible spectrophotometer (Shanghai Jing Hua Technology Instrument Co., Ltd. Shanghai, China), a DF-101S collection thermostatically controlled magnetic agitator (Gongyi Yuhua Instrument Co., Ltd. Gongyi, China), an FD-1A-50 freeze-drying machine (Beijing Boyikang Experimental Instrument Co., Ltd. Beijing, China), a JA3003B Electronic Balance (Shanghai Yueping Scientific Instrument Co., Ltd. Shanghai, China) and an RE-52A Rotary Evaporator (Shanghai Yarong Biochemical Instrument Factory, Shanghai, China).

### 3.2. Preparation of Rutin Cyclodextrin Inclusion Complexes (IC)

The preparation of the inclusion complexes was carried out using the freeze-drying method [38]. An appropriate amount of rutin was weighed and dissolved in a small amount of ethanol, and the same amounts of β-CD, HP-β-CD and DM-β-CD were weighed and dissolved in distilled water. The aqueous cyclodextrin solution was stirred using a magnetic stirrer, and the ethanol and rutin solution was slowly added to the aqueous cyclodextrin solution. The mixed solution was stirred continuously at room temperature for 24 h for the inclusion reaction. The liquid was transferred to a rotary evaporator for rotary evaporation at 40 °C to evaporate the ethanol component, and the remaining aqueous solution of the mixture was filtered through a 0.45 μm filter membrane. For the rutin/β-CD system, the filtrate was washed with anhydrous ethanol to remove the free rutin and freeze-dried to obtain the inclusion complex; the filtrate of the rutin/HP-β-CD and rutin/DM-β-CD system was freeze-dried to obtain the corresponding inclusion complexes.

### 3.3. Preparation of the Rutin–Cyclodextrin Physical Mixture (PM)

Rutin and cyclodextrin were weighed separately at a molar ratio of 1:1, ground in an agate mortar for 2 min and mixed thoroughly to obtain a physical mixture of rutin and cyclodextrin.

### 3.4. Characterization

#### 3.4.1. Fourier Infrared Spectrometry (FTIR)

FTIR measurements were conducted with a Nicolet IS50 Intelligent Fourier infrared spectrometer (Thermo Fisher Technologies, Massachusetts, USA). The FTIR spectra were obtained by scanning the specimens 32 times in the wavenumber range from 500 cm^−1^ to 4000 cm^−1^ with a resolution of 8 cm^−1^. The test samples were acquired by using ultra-thin disk specimens pressed in anhydrous potassium bromide (KBr).

#### 3.4.2. X-ray Diffraction Analysis (XRD)

XRD measurements were performed on an Ultima IV Powder Polycrystalline X-ray Diffractometer (Rigaku Corporation, Tokyo, Japan) with the following determination conditions: operating voltage 40 KV, output current 20 mA, diffraction angle scanning range of 5°~60° and scanning speed of 10°/min.

#### 3.4.3. Differential Scanning Calorimetry (DSC)

DSC measurements were characterized by a DSC822E thermal analyzer (Mettler Toledo, Zurich, Switzerland). Samples weighing approximately 10 mg and sealed in aluminum were heated from room temperature to 300 °C at a heating rate of 10 °C/min under a nitrogen atmosphere.

#### 3.4.4. Ultraviolet–Visible Spectral Analysis (UV)

A UV spectrum test was performed using by a UV1800PC UV–visible spectrophotometer (Shanghai Jing Hua Technology Instrument Co., Ltd., Shanghai, China). The sample solution was scanned on the ultraviolet spectrum with a scanning range of 200~600 nm, from which the appropriate wavelength was selected. A series of rutin–ethanol solutions were prepared, and the absorbance was measured at the maximum absorption wavelength. The standard curve of rutin was plotted by its absorbance (A) versus the concentration (C), and the standard curve equation was obtained. We created a series of aqueous solution concentrations of the cyclodextrins and stirred them in an excess quantity of rutin at room temperature for 48 h to determine the dissolving balance, filtration the appropriate filtrate dilution, and we determined the ultraviolet absorbance at 265 nm. Next, we calculated the solubility of the different concentrations of rutin in the cyclodextrin solution using the solubility of rutin as the ordinate and the cyclodextrin concentration as the abscissa to obtain the map phase solubility.

## 4. Simulation Part

### 4.1. Molecular Docking

The initial configuration of cyclodextrins and flavonoids required for the molecular docking established by the MS 2018 software and geometric optimization was carried out. The software used for the molecular docking simulation was AutoDock 4.0. Cyclodextrin, as the main body of the inclusion complex, was set as the rigid receptor molecule, and rutin, as the object of the inclusion complex, was set as the ligand molecule [32]. A semi-flexible method was adopted for the docking. During the docking process, the size of the selected grid box included the entire cyclodextrin, thus facilitating the identification of the docking site of the cyclodextrin molecules. The molecular docking process used the Lamarck genetic algorithm (LGA) to connect the flavonoids to the corresponding cyclodextrin cavity, and the subsequent conformational search was conducted 1000 times. For the 1000 inclusion complex structures obtained, the most stable one was selected based on the results of the energy cluster analysis.

### 4.2. Molecular Dynamics (MD) Simulation

All the models (the three cyclodextrins, rutin and the three inclusion compounds) were simulated by MD. First, we performed geometric optimization. We selected the COMPASS force field in the Force Calculation of Modules module. The energy optimization was 2 million steps, and the convergence value was 1.0 × 10^−5^ kcal/mol [33]. Second, we established a rutin/β-CD cell, and the number of inclusion complexes in the cells was 1–10, respectively. Two hundred cycles of unit-cell annealing were carried out after the geometric optimization, and the temperature range was from 200 K to 500 K. Then, a dynamic simulation was carried out at 298 K through NVT ensemble (constant atomic number, constant volume and constant temperature), and the simulation steps were set to 1000 ps. At this stage, the temperature simulated by NPT ensemble (constant atomic number, constant pressure and constant temperature) was 298 K, the pressure was 0.1 Mpa and the step size was 1000 ps. Finally, according to the density and the calculated solubility parameters in the results of the last frame and the comparison with the cell with only one inclusion compound, it was determined that the cell system with seven inclusion compounds was the closest; thus, the cell system of the other systems was established with seven inclusion compounds, and the other simulations and parameters were the same as those of the rutin/β-CD system. The molecular dynamics simulation could judge the system balance, including the temperature balance and energy balance.

## 5. Conclusions

In this study, inclusions of three cyclodextrins with rutin were prepared using the saturated solution method, and the structures of the inclusions were characterized by FTIR, XRD, DSC and UV–visible spectroscopy. The inclusion behaviors were studied theoretically using molecular docking and molecular dynamics simulations, and the effects of the different cyclodextrins on the inclusion of rutin were investigated. The results of the FTIR, IR, XRD and DSC analyses demonstrated that the inclusion ratio of the three cyclodextrins with rutin was 1:1, and the stability constants of the three inclusion complexes were K_S_ (rutin/β-CD) = 275.5 M^−1^, K_S_ (rutin/HP-β-CD) = 442.5 M^−1^ and K_S_ (rutin/DM-β-CD) = 1012.4 M^−1^, respectively. The stoichiometric ratio of the host and guest of the molecular docking was 1:1. The results of the solubility parameters, binding energy, MSD, hydrogen bond analysis and RDF showed that the rutin/DM-β-CD system had the best solubility and that the rutin/DM-β-CD system was the most stable inclusion complex. The experimental results were consistent with the simulation results, indicating that, among the three cyclodextrins, DM-β-CD and rutin had the best inclusion effect. The successful preparation of clathrate reduced the limitations of the application of rutin, and the application of rutin in food and pharmaceuticals will be more extensive. The study and comparison of the three cyclodextrins obtained the best inclusions, offering guidance for choosing solubilized carriers for rutin.

## Figures and Tables

**Figure 1 molecules-28-00955-f001:**
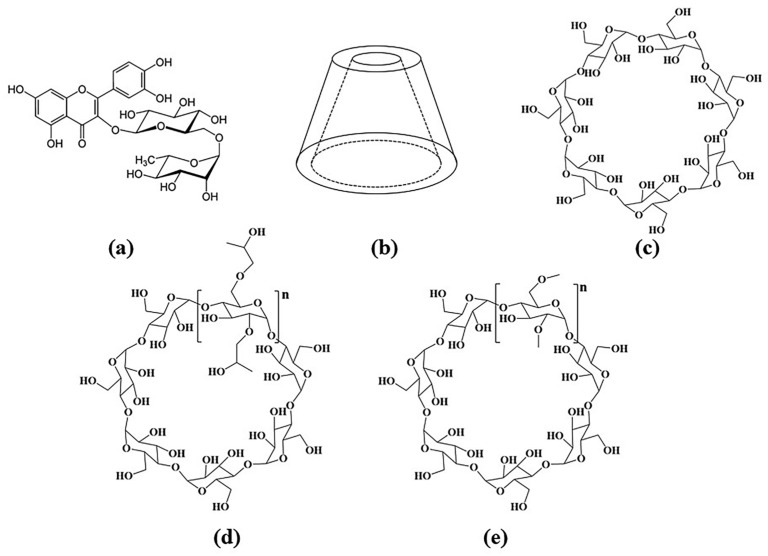
Structural formula of (**a**) rutin, and (**b**) Stereoscopic structure of cyclodextrin, (**c**) β-CD, (**d**) HP-β-CD, and (**e**) DM-β-CD.

**Figure 2 molecules-28-00955-f002:**
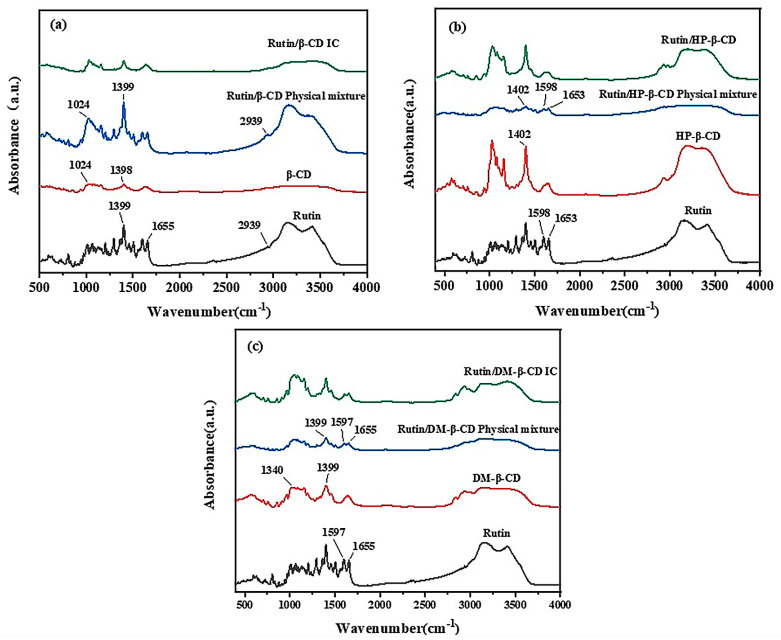
FTIR spectrum of (**a**) rutin, β-CD, physical mixture and inclusion complex; (**b**) rutin, HP-β-CD, physical mixture and inclusion complex and (**c**) rutin, DM-β-CD, physical mixture and inclusion complex.

**Figure 3 molecules-28-00955-f003:**
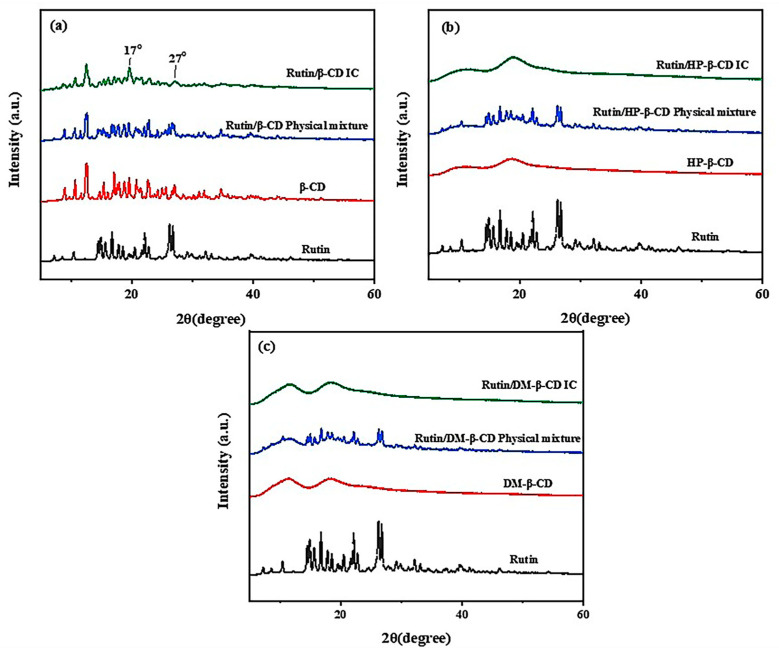
X-ray diffractometry spectra of (**a**) rutin, β-CD, rutin/β-CD physical mixture, rutin/β-CD IC; (**b**) rutin, HP-β-CD, rutin/HP-β-CD physical mixture, rutin/HP-β-CD IC; (**c**) rutin, DM-β-CD, rutin/DM-β-CD physical mixture and rutin/DM-β-CD IC.

**Figure 4 molecules-28-00955-f004:**
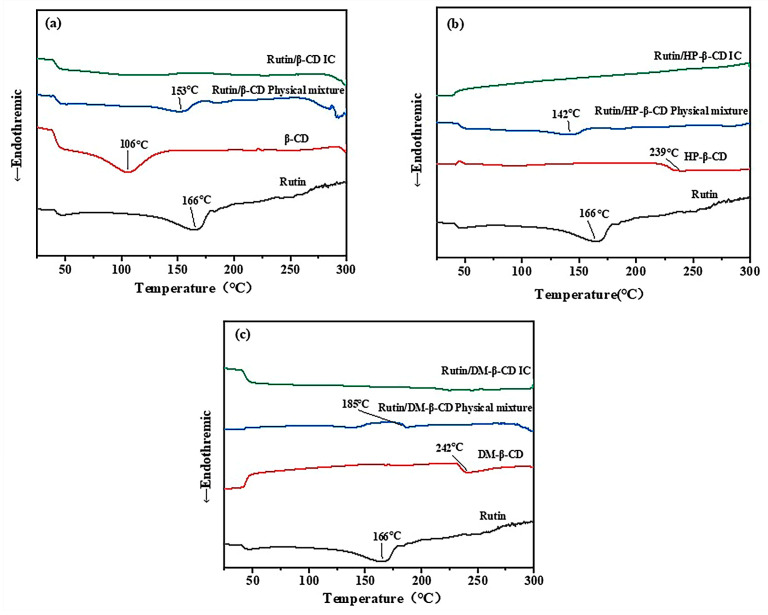
DSC curves of (**a**) β-CD, rutin, rutin/β-CD physical mixture, rutin/β-CD IC; (**b**) HP-β-CD, rutin, rutin/HP-β-CD physical mixture, rutin/HP-β-CD IC; (**c**) DM-β-CD, rutin, rutin/DM-β-CD physical mixture and rutin/DM-β-CD IC.

**Figure 5 molecules-28-00955-f005:**
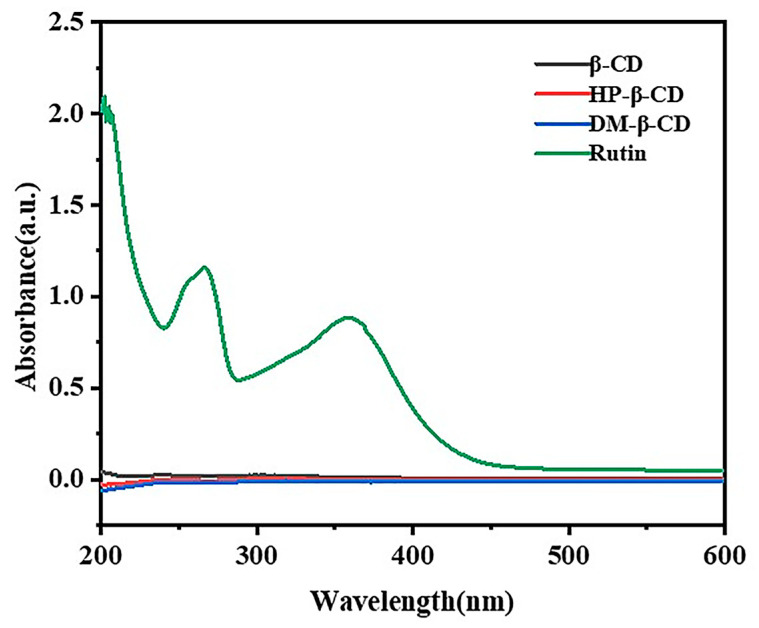
UV spectra of β-CD, HP-β-CD, DM-β-CD and rutin.

**Figure 6 molecules-28-00955-f006:**
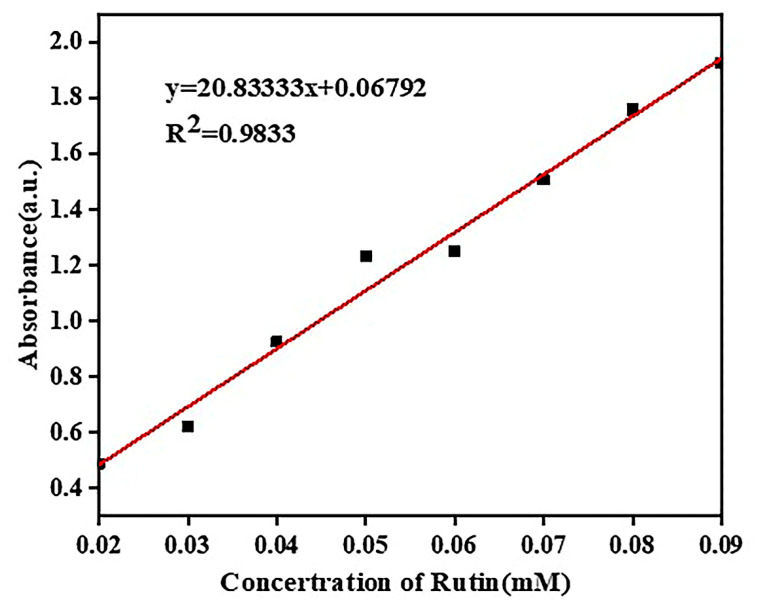
Standard curve of rutin.

**Figure 7 molecules-28-00955-f007:**
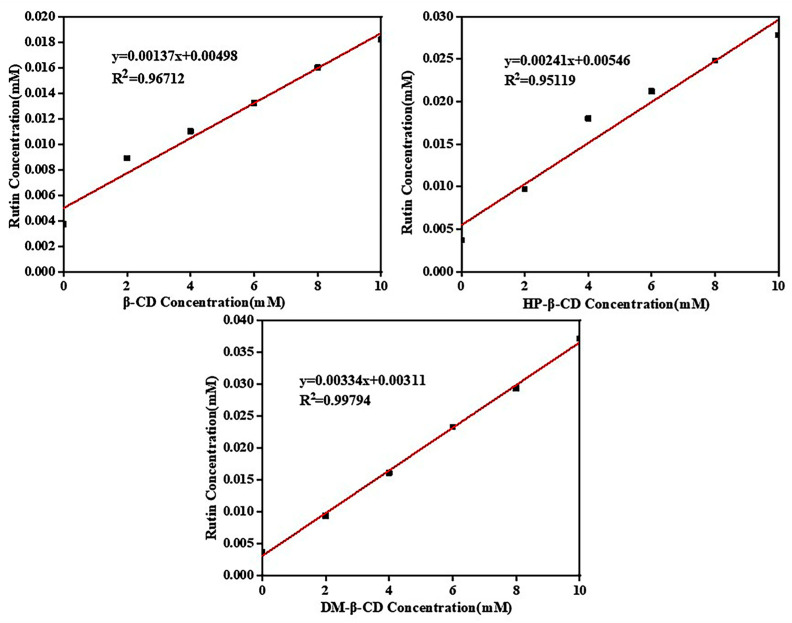
Phase solubility diagram of three cyclodextrins and rutin.

**Figure 8 molecules-28-00955-f008:**
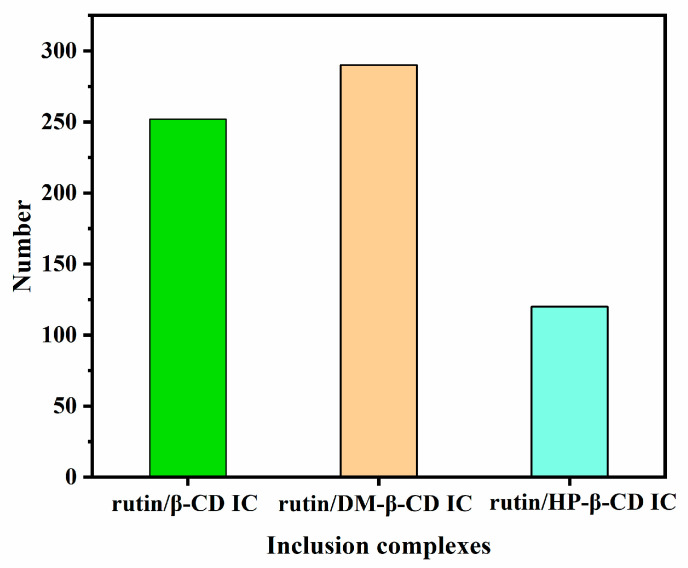
Number of images of an optimal configuration of a triple inclusion complex.

**Figure 9 molecules-28-00955-f009:**
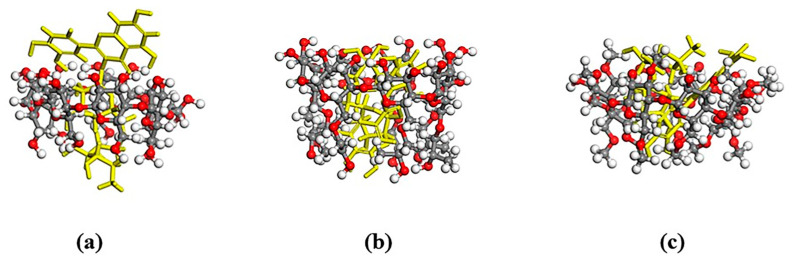
The most-stable conformations of (**a**) rutin/β-CD; (**b**) rutin/HP-β-CD and (**c**) rutin/DM-β-CD. (The yellow molecule represents rutin. The white, red and gray spheres represent the H, O and C atoms in cyclodextins, respectively.)

**Figure 10 molecules-28-00955-f010:**
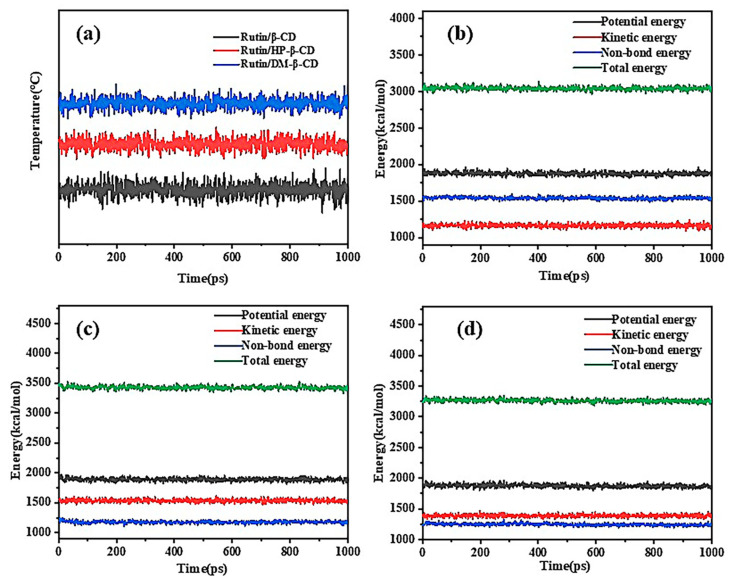
(**a**) Temperature equilibrium diagram of three inclusion complexes; (**b**) energy balance diagram of rutin/β-CD inclusion complex; (**c**) energy balance diagram of rutin/HP-β-CD inclusion complex and (**d**) energy balance diagram of rutin/DM-β-CD inclusion complex.

**Figure 11 molecules-28-00955-f011:**
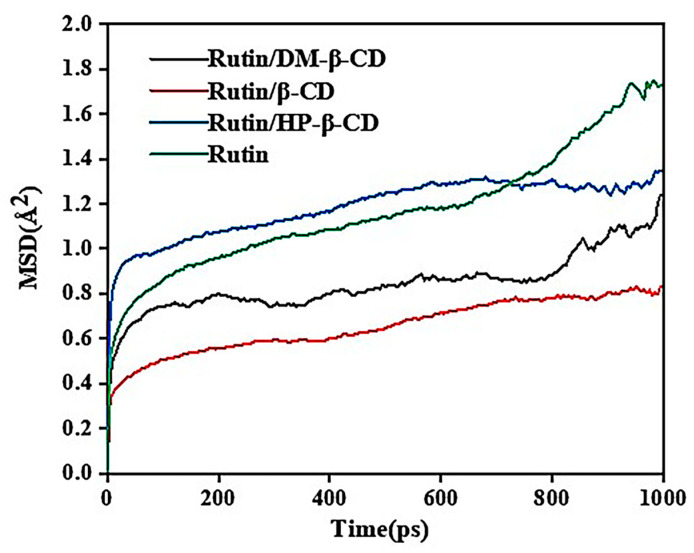
MSD curves of cyclodextrin–rutin inclusion complexes and rutin.

**Figure 12 molecules-28-00955-f012:**
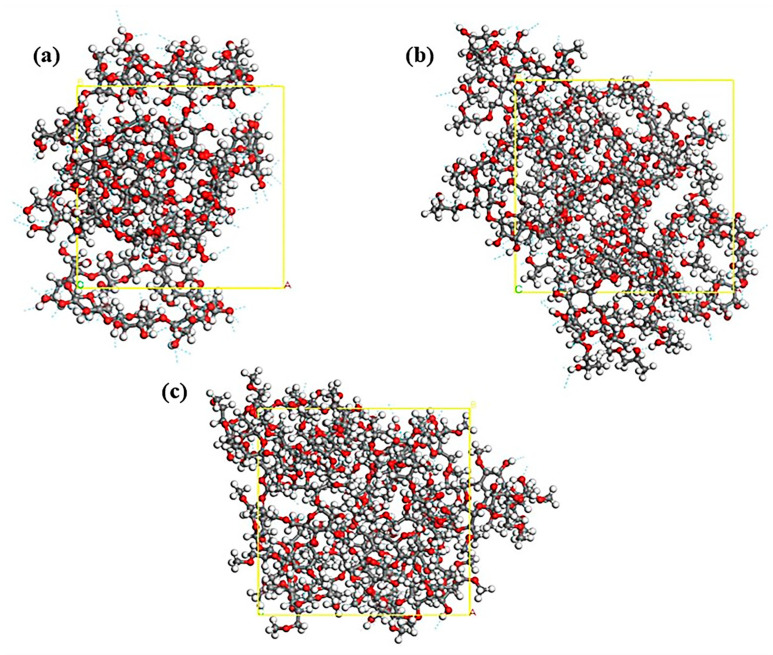
Models for MD simulation of (**a**) β-CD system, (**b**) HP-β-CD system, (**c**) DM-β-CD system (the white atom represents H, the gray atom represents C, the red atom represents O and the blue dashed line represents hydrogen bonds, the yellow squares represent the cells).

**Figure 13 molecules-28-00955-f013:**
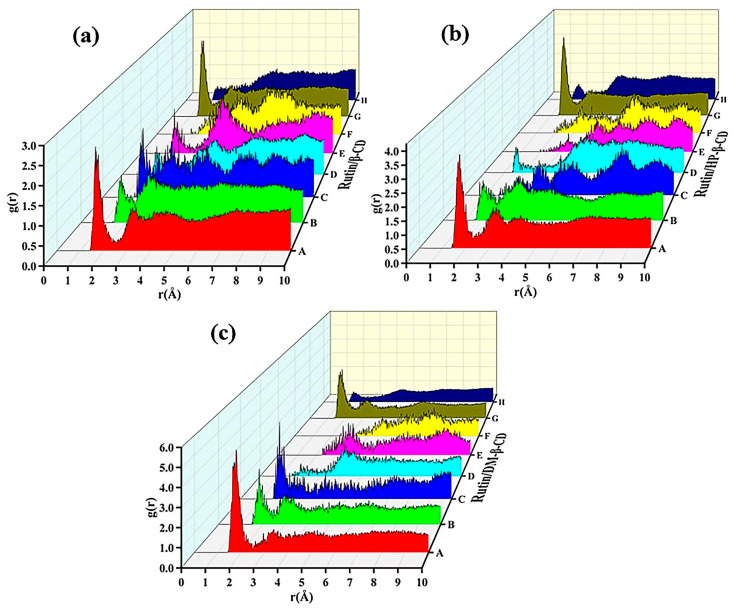
RDF of (**a**) rutin/β-CD IC, (**b**) rutin/HP-β-CD IC and (**c**) rutin/DM-β-CD IC.

**Table 1 molecules-28-00955-t001:** UV absorbance of rutin–ethanol solution with different concentrations.

Concentration (mM/L)	0.02	0.03	0.04	0.05	0.06	0.07	0.08	0.09
Absorbance	0.489	0.622	0.926	1.232	1.250	1.507	1.758	1.926

**Table 2 molecules-28-00955-t002:** Absorbance of rutin aqueous solution with different concentrations of cyclodextrin.

Concentration	0 mM/L	2 mM/L	4 mM/L	6 mM/L	8 mM/L	10 mM/L
β-CD	0.145	0.253	0.298	0.343	0.401	0.447
HP-β-CD	0.145	0.270	0.443	0.509	0.584	0.648
DM-β-CD	0.145	0.264	0.403	0.554	0.678	0.843

**Table 3 molecules-28-00955-t003:** Solubility parameters of β-CD, HP-β-CD, DM-β-CD and rutin.

*δ*_A_ (cal/cm^3^)^1/2^	*δ*_B_ (cal/cm^3^)^1/2^	δ_A–B_ (cal/cm^3^)^1/2^
*δ*_rutin_ = 12.650	*δ*_β-CD_ = 9.658	2.992 < 3.4
*δ*_rutin_ = 12.650	*δ*_HP-β-CD_ = 10.677	1.973 < 3.4
*δ*_rutin_ = 12.650	*δ*_DM-β-CD_ = 12.422	0.228 < 1.0

**Table 4 molecules-28-00955-t004:** Binding energies of the three cyclodextrin–rutin inclusions.

Complex	Binding Energy (Kcal/mol)
rutin/β-CD	111.49
rutin/HP-β-CD	114.95
rutin/DM-β-CD	519.82

**Table 5 molecules-28-00955-t005:** Number of hydrogen bonds and hydrogen bond concentration in the three cyclodextrin cells.

Cyclodextrin	Number of Cell Hydrogen Bonds	Hydrogen Bond Concentration (mol/cm^3^)
β-CD	143	1.22 × 10^−3^
HP-β-CD	144	7.92 × 10^−3^
DM-β-CD	42	2.96 × 10^−2^

**Table 6 molecules-28-00955-t006:** Atomic charge distribution of rutin and cyclodextrin.

Rutin Atoms	Electronegativity	Cyclodextrin Atoms	Electronegativity
O1	−0.57	O7	−0.57
O2	−0.452	O8	−0.32
O3	−0.419	H2	0.41
O4	−0.32		
O5	−0.1815		
O6	−0.0685		
H1	0.41		

## Data Availability

Data are contained within the article.

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
