# Peer review of "Preparation, Characterization and Molecular Dynamics Simulation of Rutin–Cyclodextrin Inclusion Complexes"

_molecules, 2023, doi:10.3390/molecules28030955_

Round 1
Reviewer 1 Report
1. Shorten the introduction section. Keep information related to study design.
2. Freeze drying condition must be elaborated with reference.
3.Phase solubility section and dissolution experimental part missing. Authors have added the result part of many sections but experimental section missing.
4. NMR is missing. IR does not confirm the formation of complex.
5. Conclusion and abstract must contain important findings of the study.
Reviewer 2 Report
The study of Chang, et al., developed Rutin cyclodextrin inclusion complexes by encapsulation of rutin with three different cyclodextrins mainly beta-cyclodextrin (β-CD), 2-hydroxypropyl beta-cyclodextrin (HP-β-CD) and 2,6-dimethyl beta-cyclodextrin (DM-β-CD) using the freeze-drying method and further analyzed using a combination of experimental and simulation methods to characterize and demonstrate the formation of the inclusion complexes. The experimental and simulation results also confirm that the inclusion effect of DM-β-CD and rutin is consistent and best. Although the manuscript is organized and has the researcher's interest. The methodology adopted is quite satisfactory. However, some caveats must be addressed before the study deserves publication.
1. As authors have reported many papers related to rutin-cyclodextrin inclusion in the introduction and a number of papers have also already been published. So, the authors should improve the introduction part by comparing the rationale of this developed rutin inclusion versus reported/published in the literature.
2. Authors should also include a few advantages or disadvantages of this inclusion complex after comparing it with other reported formulations for rutin such as the rutin phospholipid complex by Hafsa et al., 2016 and nanoformulations by Federica et al., 2021 and so on. Without knowing this, it is not possible to assess the novelty, and potential impact, of this paper. Also, justify how this formulation is more suitable rather than other developed formulations.
3. Authors should mention the ratio or precise quantity of rutin, and various cyclodextrin used in the preparation of inclusion in the method sections.
4. Authors should double-check the manufacturer name of instruments used in the method section which should be incorporated in the revised manuscript.
5. Was the physical stability as well as light stability of the developed inclusion checked as rutin has photolytic degradation behaviors? Please explain.
6. It would be advantageous to include more experimental characteristics about prepared inclusion complexes by thermodynamic solubility and in vitro release behavior as compared with the bare drug.
7. Reviewers encourage to authors if they include some in vivo Pharmacokinetics experiments in a rat model to strengthen the outcomes of the present study.
8. Authors should compare the results of Tables 3 and 4 by statistical program and include them in the manuscript.
9. Authors should improve the figure DPI quality and size in the revised manuscript.
10. Authors should elaborate more about the future perspective of this inclusion complex in the conclusion section.
11. There are minor grammatical and typo errors throughout the manuscript which should be corrected.
Round 2
Reviewer 1 Report
Accept
Reviewer 2 Report
Authors addressed the all comments and made changes to the manuscript accordingly. I support that the manuscript can be accepted for publication in Molecules.